# Influence of Pastoral Settlements Gradient on Vegetation Dynamics and Nutritional Characteristics in Arid Rangelands

Akash Jamil [1], Muhammad Zubair [1,*] and Bryan A. Endress [2,*]

1  Department of Forestry and Range Management, Bahauddin Zakariya University, Multan 66000, Pakistan; akashjamil2@live.com
2  Eastern Oregon Agriculture Research Center—Union Experiment Station, Oregon State University, La Grande, OR 97850, USA
*  Correspondence: zubair.fast@bzu.edu.pk (M.Z.); bryan.endress@oregonstate.edu (B.A.E.)

**Abstract:** An expansion of pastoral settlements in arid rangelands can increase pressure on fragile ecosystems. High stocking rates and inappropriate rangeland management can cause large, difficult-to-reverse changes in the composition and structure of rangeland ecosystems. This study aims to understand changes in vegetation composition (cover, density, biomass, richness, diversity) and nutritional characteristics of range vegetation along a gradient of increasing pastoral population in arid rangelands of Pakistan. Three sites were selected for sampling within three areas classified by their distance from settlement: Near, Away, Far (9 sites in total) belt transects (10 × 150 m). A total of 90 transects (30 each in classified sites) of size 10 × 150 m were placed at a distance of 100 m apart from each distance group. Results identified 28 species from nine plant families. We found a significant difference in vegetation characteristics along the gradient. Vegetation diversity increased along the settlement gradient, highest being in the far sites. Density, plant cover and biomass were greatest at the sites near pastoral communities and decreased as the distance from the settlements increased. Palatable species were characterized by low density and diversity near herder's villages, but values increased as the distance from the villages increased. An increase in shrubs was observed near pastoral settlements, resulting in higher plant cover, biomass and density in these sites. This study identified anthropogenic impacts on rangeland structure and composition and found large shifts near communities. The frequent monitoring of vegetation resources is important, and the development of sustainable conservative strategies are recommended to ensure harmonious coexistence of pastoral populations in arid rangelands.

**Keywords:** range vegetation; diversity; rangeland ecology; vegetation inventory; population gradient; nutrient analysis; veg measure; range monitoring; pastoral settlements

## 1. Introduction

Pastoralism in tropical and arid rangelands is one of the oldest livelihoods centered on animal husbandry [1]. Around 500 million people globally are directly or indirectly involved with livestock rearing and its production [2]. An increasing population and rapid conversion into agricultural lands have expanded the pastoral population in most underdeveloped regions. For example, the population of pastoral families since the 1990's in Asia alone has reached ~50 million [3]. It is reported that the increased pastoral settlements in the past few decades have also enhanced their spread deeper into uninhabitable arid rangelands [4]. This has also led to a higher number of livestock acquired by each pastoral family in the settlements [5]. The number of livestock heads in south Asia alone has quadrupled to 140 million heads in 2019 [6].

Studies have shown that about 96% of natural rangelands are now inhabited by 420 million pastoral settlements [5,7]. These pastoral communities range from limited rotation to transhumance and nomadic in terms of mobility [8]. Pastoralists are continuously

on the move; thus, they are able to access new pasture and harvest a good quantity and quality of forage [9]. An increasing livestock population and continuous movement of herders negatively affects standing biomass and lower plant diversity, that is leading many rangelands in South Asia and Africa towards desertification [10,11]. This is mostly due to a decline in seed production and seed number in the soil, resulting in lower regrowth levels over the long term [12]. Increased stocking rates in this part of the world have been identified as unsustainable and are thought to be the cause of declining net primary productivity of range resources [13].

Pastoral resource-use patterns are studied using various mobility models [14,15]. According to these models, forage distribution and daily herding management determine grazing pressure distribution [16,17]. Researchers' argue that grazing intensity is uniform within a specified distance from a herder's settlement [18]. However, studies have shown grazing intensity decreases as the distance from the settlement is increased. Locations near pastoral settlements are termed as sacrificial zones due to recursive livestock use while the regions farther from pastoralists are known as transition zones and are rarely subjected to livestock grazing [19,20]. However, a recent study conducted in the arid rangelands of southern Pakistan depicted heavy grazing in areas far from pastoral settlements. This study found that regions further away from the settlements had lower plant density and sparse vegetation cover compared to nearer sites [21].

Empirical research suggests that pastoralists move far beyond their settlement areas in search of greener and good quality forage. This can alter the natural composition of herbaceous vegetation by reducing the abundance of perennials [13]. Heavy grazing on perennial species can alter competitive interactions between palatable perennial species and lower-palatable annual species [22]. This causes a shift in vegetation and can result in the dominance of unpalatable and grazing-tolerant annual species [21].

Trampling, grazing and excreta deposition not only affects vegetation, but also alters soil nutrient dynamics. Various studies show heavily grazed rangelands have a lower quality and quantity of soil nutrients [23–25]. Heavy grazing affects the accumulation of carbon (C), nitrogen (N), phosphorous (P) and potassium (K) in the soil due to modification of N and C cycles [23]. Rangeland degradation has been both a cause and a consequence of decreasing soil macro- and micronutrients. Diminishing soil nutrients amidst degrading arid lands is frequently reported in South Asia [26–28] and globally [8,29,30]. Many studies have focused on the alteration of micro and macro nutrient status in grazing lands amidst increasing and expanding pastoral settlements in arid rangelands [31,32].

Various studies have employed vegetation analysis on rangeland ecosystems to assess and characterize anthropogenic degradation [33,34]. Information regarding floristic composition, biomass and vegetation cover and the effect of grazing on vegetation are perquisites for the development of sustainable management plans and grazing systems [35,36]. Floristic analysis associated with camps, rest areas and watering points is a commonly used approach for assessing the intensity of grazing [33]. These factors can accurately monitor the dynamics of the vegetation and its extent of change due to anthropogenic disturbances [37].

While concerns have been raised about the effects of the increased pastoralists higher livestock densities in arid ecosystems, accurate information that characterizes the changes to vegetation amidst this expansion is limited [35–37]. Specific changes to vegetation characteristics have yet to be fully explored. In arid rangelands, natural vegetation plays a key role in the functioning of the ecosystem, sustaining of the biodiversity and productivity of the wildlife and human communities [35]. It is then extremely important to understand and monitor the dynamics and change in vegetation characteristics amidst the expansion of pastoral settlements. In a Pakistani context, research is of much need due to an increasing population and expansion of pastoral settlements in the rangelands throughout the country.

This study aims to provide pivotal information regarding the effect of pastoral populations on the vegetation cover, the diversity of palatable species and the mineral composition of forage resources along pastoral settlements in the Thal desert of Pakistan. The area is undergoing an increase in pastoralist settlement, and there is concern about the associated

impact to rangelands of the region. This knowledge is essential for meaningful collaboration between government departments and local pastoral communities, and information on the situation can provide important baseline information to develop sustainable management practices in this important ecosystem.

The main objectives of this study are: (1) to characterize the diversity, composition, forage and biomass of vegetation in the region and (2) assess any association between vegetation characteristics and distance from pastoral settlements. We hypothesized that plant communities near pastoral settlements will have lower plant species diversity, biomass and density of preferred forage species and greater abundance of toxic and unpalatable species compared with plant communities further away. We also hypothesized that distance from the pastoral settlements will have a positive effect on the nutritional quality of grazing vegetation.

## 2. Materials and Methods

### 2.1. Site Description & Environmental Conditions

The present study was conducted in rangelands of Thal desert situated in district Layyah (Figure 1). It is located between 31°10′ N and 71°30′ E in southern Punjab, Pakistan. A subtropical sandy desert that is bounded by Indus River at its west while in north is Jhelum and Chenab river flood plains lies on its east [36]. Thal desert is the third largest desert in the country, occupying 2.5 million hectares. It is characterized by semi-arid to hyper-arid climatic conditions, having rainfall less than 200 mm. Most of this rainfall occurs in the monsoon season between the middle of June and late August (450–800 mm). These regions are often quite hot, experiencing temperatures that rise until 46 °C in the summer while in the winters, they can fall to 1 °C in the nights.

Range Resources and Pastoral Settlements of Thal Desert

Most of the desert is characterized by dry sandy plains with extensive sand dunes that shift with strong winds. Thal consists of 32% grazing lands. These grazing resources are divided into 9 Rakhs (Range management units) controlled by the forest and range management department Punjab. These rangelands harbor some of the most nutritious herbs, grasses, shrubs and woody vegetation. These species have been sustaining the livelihood of various pastoral settlements for centuries.

These villages traditionally are famous in harboring pastoral settlements living in these lands, with traditional herders raising and feeding livestock for generations. According to the Pakistan Bureau of Statistics, five villages include 1080 households that are completely dependent upon natural forage resources for their daily subsistence and income generation. The grazing system prevalent in these regions is mostly year-round continuous grazing system at fixed prices determined by the forest range department of Punjab.

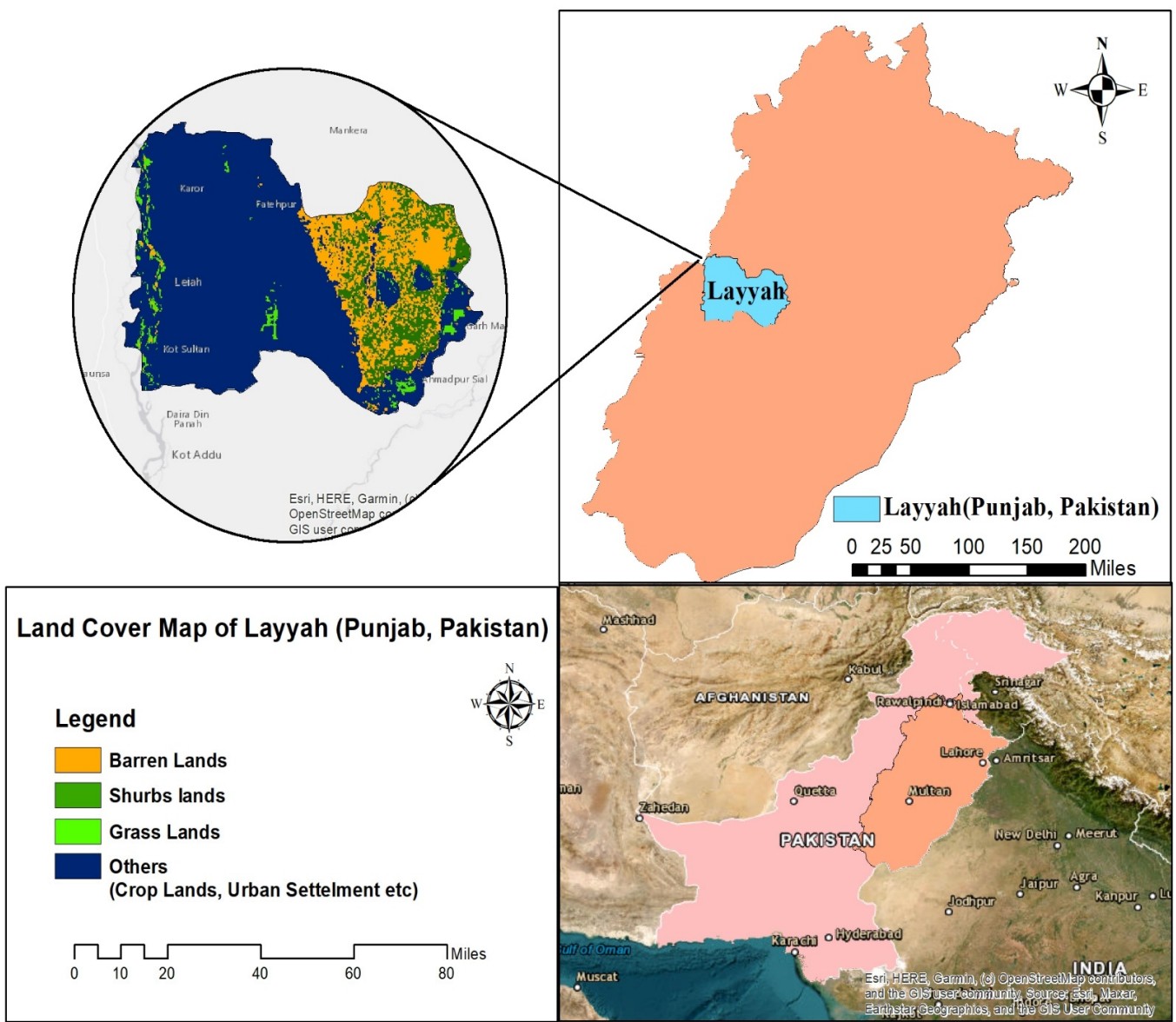

**Figure 1.** Map of the Study site (District Layyah, South Punjab).

### 2.2. Site Selection and Vegetation Sampling

We measured the impact of pastoral settlements on (1) vegetation diversity, (2) vegetation density, (3) vegetation cover (4) biomass and (5) nutritional characteristics of vegetation in Rakh Khairaywala. Three sampling sites were established in three regions classified by distance from settlements: Near (1–2 km from settlements), Away (2–4 km) and Far (4–6 km). The sites near settlements had somewhat scattered clusters of pastoral households. The sites away from settlements had sparse presence of pastoralists' houses and livestock enclosures while the sites far from settlements had no sign of any households.

Three sites were selected in each of the 3 classified areas (Near, Far, Away) for a total of 9 sites. Site selection was based on professional judgement, experience and skill of the observers combined with the usage of vegetation maps, photos, field information and ground truthing [37].

At each of the 9 sites, we established 10, 150 m long transect spaced 100 m apart. On each transect six, 1 m$^2$ plots were established at 30 m intervals, resulting in 60 plots per site and 180 plots sampled within each of the three distance categories. Overall, we sampled

vegetation in 540 plots. Sampling was carried out when plants were mature during post monsoon season at forage's peak biomass.

We used quadrat estimation method to determine species composition, cover, biomass and diversity. We clipped all vegetation within each plot at 2.5 cm from the base of the plant. Fresh weights of the clipped samples were taken utilizing a portable electric weigh balance and were then placed in paper bags. The paper bags were transported back to the laboratory and placed into a preheated oven at 70 °C for 48 h or until a constant weight was attained. The biomass of the samples was calculated using a balance. Local grazers and experienced elders in the settlements helped with the identification of plant species. The nomenclature of these species was based on the global biodiversity information facility (gbif.org).

### 2.3. Biodiversity Analysis

In order to analyze biodiversity of available forage along the gradient of pastoral settlements, we calculated several metrics. First, we calculated Frequency (F), Density (D) and Abundance based on our estimates of percent cover within the quadrats (plots). The species abundance and distribution were obtained from Miller and Wiegert [38]. We analyzed dominance of species among each site using Important value index (IV). In order to analyze species richness, evenness, heterogeneity and dominance, we utilized Menhinick Index, Buzas and Gibson, Shanon and Simpson diversity indices respectively.

#### 2.3.1. Vegetation Palatability Preference

Knowledge regarding palatability preference of livestock were obtained from experienced pastoralists. Among various plant species, the herders were allowed to categorize plants based on highly palatable, palatable, less palatable and unpalatable. Then, all of this information was confirmed utilizing a dedicated study on fodder palatability in the Thal rangelands [36].

#### 2.3.2. Vegetation Cover Determination

The canopy cover of the range vegetation was determined using Digital vegetation charting technique (DVCT). Radom pictures for ground cover estimates along all the classified sites were taken from 1.5 m from the ground using a high-resolution digital camera according to the protocols of DVCT mentioned in Louhaichi et al. [39]. Nikon Coolpix Aw125 equipped with a 28-mm lens was used for random photography of the range vegetation in the sampling sites. The dimension of each photograph were $4384 \times 3188$ pixels. The size of each picture was 3641 kb in JPG format.

The estimation of green vegetation cover percentage was calculated from the Images via a supervised classification technique using VegMeasure software® [39]. VegMeasure is a technique that allows vegetation measurement through non-destructive approach. This software interprets the color from the images taken from the digital camera to formulate meaningful classes and allows the customization of images. The current study only utilized two classes (i.e., vegetation cover and bare ground or soil).

### 2.4. Forage Quality: Sample Preparation and Analysis

The samples for micro- and macro-mineral analysis were collected randomly along transects of every plant species encountered during the survey. After collecting the plant samples, they were washed with de-ionized water and then were dried in the oven for two days at 80 °C. It was then subjected to crushing in order to make a powder. About 2 g from each plant sample in the powder form were dissolved in 10 mL of $HNO_3$ for about 12 h. It was then heated until reddish brown fumes disappeared. The solution was then added in to 4 ml of per chloric acid and was heated for about 5 min. After adding water, the solution was heated up until marked volume of 250 mL was reached. The analysis of micro- and macro-mineral content analysis was conducted in Atomic absorption spectrophotometer using flame atomic absorption spectroscopy utilizing Perkin Elmer A Analyst 700.

*2.5. Statistical Analysis:*

The data was statistically analyzed using SPSS (v 26, IBM Corp., Armonk, NY, USA) and R (v 3.3.2, R Foundation for Statistical Computing, Vienna, Austria). In order to test the skewness of the data, we employed Shapiro–Wilk's test that depicted the data to be negatively skewed. This then lead to the utilization of the non-parametric test Kurskal–Wallis *H* test to determine statistically significant differences between more than two groups. This test aimed to estimate if there was significant variation in vegetation characteristics along distance gradient from pastoral settlements.

## 3. Results

*3.1. Floral Characteristics:*

The survey identified that the region was characterized by sparse vegetation. We reported a total of 28 plant species from 9 different families. The rangeland from which samples were obtained used to be open for grazing year-round. All of the plant species identified in this survey were endemic to this region. Nearly 61% of the vegetation recorded were grasses, 24% of the plants were shrubs and 12% were herbs (forbs). Just 3% of the plant species were ephemerals forbs.

About 90% of the plant species found in this rangeland were used as forage resource for livestock. The vegetation survey was conducted along the distance gradient from pastoral settlements. The number of species in the sites near settlements was low, but the number of individual plants was greater compared to other sites. The sites far from communities displayed higher diversity but low density compared to other two sites.

The results showed that unpalatable species, particularly toxic shrubs, dominated the areas near the pastoral settlements. Shrubs like *Aerva javanica* (Burm.f.) C.Jussieu ex Schult, *Kochia indica* Wight, *Haloxylon salicornicum* (Moq.) Bunge and *Calotropis procera* (Aiton) Aiton fil. had higher densities and plant cover. The rapid growth of these shrubs not only hampered the growth of palatable species, but also decreased the diversity of vegetation. However, sites away and far from pastoral settlements displayed higher plant diversity, especially in terms of palatable forage. These sites also had lower plant densities compared to the other two sites. It was mainly due to the grazing pattern, behavior and routes of the local herders. Sites away from pastoral settlements depicted a moderated number of plant individuals and species diversity.

*3.2. Family Distribution*

A total of 9 families were encountered in the surveys. The most dominant and abundant family was the Poaceae. The data revealed more than 500 individuals of this particular family. The second most dominant family was the Amranthaceae. The least abundant family was Tetradiclidaceae. Important value index depicted no equal distribution of dominant species in all of the study area.

The results showed a significant relationship between the plant density and the distance from pastoral settlements (Tables 1 and 2). It was observed that the vegetation density had highly significant difference with the distance of pastoral communities (Tables 1 and 2). The mean values showed larger plant densities in the sites near herder communities, and as the distance would increase, the plant population decreased (Table 1).

**Table 1.** Descriptive statistics of vegetation characteristics.

| | | Mean ± SE Near | Range (Min–Max) Near Sites | Median | Mean ± SE Away | Range (Min–Max) Away Sites | Median | Mean ± SE Far | Range (Min–Max) Far Sites | Median |
|---|---|---|---|---|---|---|---|---|---|---|
| Vegetation Characteristics | Biomass (kg/ha) | 19.05 ± 1.87 | 9.00–36.00 | 17.00 | 14.85 ± 1.86 | 5.00–29.00 | 10.50 | 12.10 ± 1.39 | 5.00–23.00 | 9.50 |
| | Density (/100 m$^2$) | 62.54 ± 14.00 | 10.00–203.00 | 57.00 | 45.20 ± 13.73 | 10.00–206.00 | 24.00 | 23.79 ± 4.55 | 4.00–51.00 | 16.50 |
| | Plant cover (%) | 25.15 ± 1.33 | 16.00–35.00 | 24.00 | 19.25 ± 3.26 | 4.00–57.00 | 14.00 | 15.70 ± 1.65 | 5.00–29.00 | 17.00 |
| Macro Nutrients | Na | 250.55 ± 22.02 | 157.78–387.78 | 234.44 | 301.38 ± 11.86 | 230.12–394.44 | 289.01 | 223.50 ± 12.67 | 164.44–301.09 | 216.44 |
| | Ca | 3123.9 ± 123.2 | 2502.0–3867.0 | 3127.00 | 2630.8 ± 118.8 | 1682.0–3292.0 | 2604.50 | 1847.4 ± 64.1 | 1432.0–2167.0 | 1844.50 |
| | K | 812.5 ± 54.3 | 556.0–1096.0 | 888.00 | 1162.9 ± 57.4 | 876.0–1576.0 | 1099.00 | 594.7 ± 30.8 | 414.0–765.0 | 580.00 |
| | Fe | 68.95 ± 3.12 | 30.18–58.36 | 50.55 | 53.81 ± 3.37 | 43.13–79.27 | 60.18 | 39.29 ± 1.56 | 38.36–79.28 | 64.20 |
| | Zn | 102.18 ± 2.45 | 56.14–86.29 | 73.29 | 75.40 ± 2.39 | 58.12–85.43 | 78.29 | 85.00 ± 2.64 | 68.29–100.43 | 84.72 |
| Micro Nutrients | Cu | 3.04 ± 0.15 | 2.16–3.88 | 2.29 | 4.12 ± 0.12 | 3.45–5.02 | 3.19 | 3.50 ± 0.17 | 2.02–4.21 | 3.28 |
| | Co | 2.49 ± 0.15 | 1.39–3.39 | 2.56 | 3.17 ± 0.18 | 2.12–4.56 | 3.23 | 2.66 ± 0.12 | 3.23–5.56 | 3.54 |
| | Mn | 2.44 ± 0.12 | 1.82–2.97 | 2.42 | 4.08 ± 0.37 | 2.08–6.64 | 4.11 | 4.69 ± 0.15 | 3.69–6.12 | 4.67 |

**Table 2.** Results of the Kruskal–Wallis test used to examine changes in vegetation characteristics along a settlement gradient.

| | | Kruskal Wallis Test | |
|---|---|---|---|
| | | Chi-Square | *p*-Value |
| Vegetation Characteristics | Biomass (kg/ha) | 8.74 | 0.013 |
| | Density (/100 m$^2$) | 7.63 | 0.022 |
| | Plant cover (%) | 11.89 | 0.003 |
| Macro Nutrients | Na | 11.99 | 0.002 |
| | Ca | 26.20 | 0.0012 |
| | K | 26.36 | 0.009 |
| | Fe | 24.99 | 0.006 |
| | Zn | 25.03 | 0.000 |
| Micro Nutrients | Cu | 18.36 | 0.000 |
| | Co | 6.35 | 0.042 |
| | Mn | 20.07 | 0.000 |

*3.3. Near Pastoral Settlements*

Range vegetation near pastoral settlements was mostly dominated by toxic unpalatable plant species (Figure 2). Palatable grasses and shrubs in these sites were minimal, depicting low diversity. However, study sites near pastoral settlements showed lower diversity, but the density of unpalatable species was higher (Table 1). The most dominant unpalatable species recorded in these study sites was *Aerva javanica* (Burm.f.) C.Jussieu ex Schult (Frequency (F) = 43% Density (D) = 224) *Kochia indica Wight* (F = 32% D = 156) appeared to be the second most abundant unpalatable species that was rapidly growing in these sites (Table 3).

There were quite a few palatable species in the areas near pastoral settlements, but the density of these species were less compared to unpalatable species. The most common palatable species found here were *Cynodon dactylon* (L.) Pers. (F = 38% D = 198) and *Cenchrus cilliaris* L. (F = 29% D = 112). The frequency and density of these palatable species were quite low and were surrounded by dense unpalatable fast-growing shrubs. The results showed that highly palatable forage species, such as *Salsola baryosma* (Roem. & Schult.) Dandy (F = 15% D = 87) and *Lasiurus scindicus* Henrard (F = 12% D = 101), had low frequency and density (Table 3). Generally, it was observed that sites near settlements had a lower diversity but a high density of unpalatable species shows intensive grazing activities in sites near pastoral settlements (Figures 1 and 2).

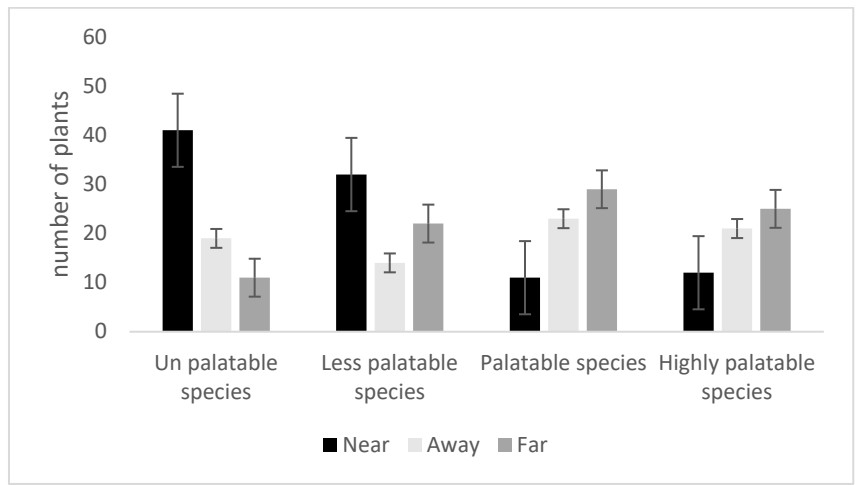

**Figure 2.** Palatable species status along settlement gradient (Unpalatable species: Aerva javanica, Calotropis procera, Heliotropium crispum, Kochia indica, Datura innoxia, Physalis divaricate and Heliotropium crispum; Less palatable species: Launaea nudicaulis, Indigofera argentea, Haloxylon salicornicum, Desmostachya bipinnata, Peganus mharmala and Poa annua; Palatable Species: Cynodon dactylon, Salsola baryosma, Leptadenia pyrotechnica, Cyperus alopecuroides, Themeda triandra, Eragrostis minor and Cymbopogon jwarancusa. Highly palatable Species: Lasiurus scindicus, Cenchrus biflorus, Salsola baryosma and Cenchrus ciliaris).

**Table 3.** Species Importance Index Values for the near human settlements in Rakh Khairawala (R.F: relative frequency, R.D: relative density, R.C: Relative Cover and I.V Species Importance Value).

| Species | Family | Classification | R.F | R.D | R.C | I.V |
|---|---|---|---|---|---|---|
| *Cenchrus ciliaris* L. | Poaceae | Highly palatable Forage | 0.27 | 0.21 | 0.15 | 0.64 |
| *Cynodon dactylon* (L.) Pers | Poaceae | Palatable forage | 0.25 | 0.58 | 0.14 | 0.98 |
| *Citrullus colocynthis* (L.) Schrader | Cucurbitaceae | Palatable (only fruits) | 0.10 | 0.030 | 0.09 | 0.23 |
| *Aerva javanica* (Burm.f.) C.Jussieu ex Schult | Amaranthaceae | Unpalatable/Toxic Shrub | 0.11 | 0.028 | 0.14 | 0.28 |
| *Salsola baryosma* (Schult.) Dandy | Amaranthaceae | Palatable Herb | 0.01 | 0.005 | 0.04 | 0.06 |
| *Leptadenia pyrotechnica* (Forssk.) Decne. | Apocynaceae | Palatable Shrub | 0.01 | 0.009 | 0.04 | 0.06 |
| *Launaea nudicaulis* (L.) Hook.fil. | Asteraceae | Less Palatable herb | 0.05 | 0.03 | 0.05 | 0.14 |
| *Calotropis procera* (Aiton) Dryand. | Apocynaceae | Unpalatable/Toxic Shrub | 0.01 | 0.003 | 0.02 | 0.04 |
| *Heliotropium crispum* Desf. | Boraginaceae | Unpalatable herb | 0.01 | 0.02 | 0.05 | 0.09 |
| *Indigofera argentea* Burm.f. | Fabaceae | Less palatable | 0.01 | 0.007 | 0.02 | 0.05 |
| *Kochia indica* Wight | Amaranthaceae | Unpalatable shrub | 0.02 | 0.06 | 0.04 | 0.23 |
| *Datura innoxia* Mill. | Solanaceae | Unpalatable shrub | 0.14 | 0.32 | 0.11 | 0.31 |
| *Lasiurus scindicus* Henrard | Poaceae | Highly Palatable Forage | 0.03 | 0.002 | 0.05 | 0.03 |
| *Physalis divaricata* D.Don | Solanaceae | Unpalatable shrub | 0.24 | 0.20 | 0.12 | 0.14 |

*3.4. Away from Pastoral Settlements*

Compared to sites near to pastoral settlements, sites away from pastoral settlements had a slightly higher representation of palatable species (Figure 2). Palatable species in these sites had higher densities, and their diversity was more than the previous sites in the survey (Table 2). Palatable species were quite diverse in sites away from pastoral settlements compared to sites near pastoral settlements. The important value index showed that the most dominant forage species in these sites was *Cynodon dactylon* (L.) Pers, having highest frequency and density (Table 4).

There was intermittent representation of quite a few moderately palatable species among them; the most notable were *Leptadenia pyrotechnica* (Forssk.) Decne. (F = 35% D = 187), *Themeda triandra* Forssk. (F = 29% D = 153) and *Eragrostis minor* Pamp. Host (F = 25% D = 110). Unpalatable species had fair representation in this site, but its diversity and density both were slightly lower compared to the previous site. A similar number

of unpalatable/toxic species were found in this site. The most abundant among them was *Aerva javanica* (Burm.f.) C.Jussieu ex Schult (F = 39% D = 198). In contrast to the first site, the density of both palatable and unpalatable species were found to be slightly lower. Generally, in this site, the density of plant species also had higher values. However, the unpalatable species depicted less density and diversity compared to near sites but still had considerable presence (Table 1 and Figure 2).

**Table 4.** Species Importance Index Values for the site away from human settlements in Rakh Khairaywala (R.F: relative frequency, R.D: relative density, R.C: Relative Cover and I.V: Species Importance Value).

| Species | Family | Classification | R.F | R.D | R.C | I.V |
|---|---|---|---|---|---|---|
| *Cynodon dactylon* L. Pers | Poaceae | Palatable Forage | 1.18 | 0.69 | 0.22 | 1.87 |
| *Cenchrus ciliaris* L. | Poaceae | Highly palatable Forage | 0.52 | 0.24 | 0.15 | 0.77 |
| *Aerva javanica* (Burm.f.) C.Jussieu ex Schult | Amaranthaceae | Unpalatable/Toxic Shrub | 0.10 | 0.01 | 0.10 | 0.12 |
| *Salsola baryosma* (Schult.) Dandy | Amaranthaceae | Highly palatable Forage | 0.03 | 0.004 | 0.06 | 0.04 |
| *Lasiurus scindicus* Henrard | Poaceae | Highly Palatable Forage | 0.03 | 0.003 | 0.05 | 0.03 |
| *Cenchrus biflorus* Roxb. | Poaceae | Highly palatable Forage | 0.09 | 0.01 | 0.10 | 0.10 |
| *Heliotropium crispum* Desf | Boraginaceae | Unpalatable herb | 0.05 | 0.004 | 0.05 | 0.059 |
| *Kochia indica* Wight | Amaranthaceae | Unpalatable shrub | 0.07 | 0.007 | 0.09 | 0.08 |
| *Haloxylon salicornicum* ( Moq.) Bunge | Amaranthaceae | Less palatable Shrub | 0.03 | 0.005 | 0.058 | 0.04 |
| *Leptadenia pyrotechnica* ( Forssk.) Decne. | Apocynaceae | Palatable shrub | 0.03 | 0.006 | 0.068 | 0.04 |
| *Cyperus alopecuroides* Rottb. | Cyperaceae | Palatable forage | 1.18 | 0.69 | 0.22 | 1.87 |
| *Themeda triandra* Forssk | Poaceae | Palatable forage | 0.97 | 0.45 | 0.34 | 1.77 |
| *Pycreus flavidus* (Retz.) T.Koyama | Cyperacea | Palatable forage | 1.1 | 0.09 | 0.07 | 1.32 |
| *Eragrostis minor* Host | poaceae | Palatable forage | 0.78 | 0.32 | 0.12 | 1.23 |
| *Panicum antidotale* Retz. | Poaceae | Palatable forage | 0.02 | 0.01 | 0.04 | 0.08 |
| *Cymbopogon jwarancusa* ( Jones)Schult. | Poaceae | Palatable forage | 0.06 | 0.04 | 0.08 | 0.19 |

*3.5. Far from Pastoral Settlements*

Sites far away from pastoral settlements had the greatest number of plant species. Plant density was minimal in this site comparatively. The number of palatable fodder species was also observed to be higher. Highly palatable species, such as *Cenchrus ciliaris* L. (F = 40% D = 181), *Cynodon dactylon L. Pers* (F = 36% D = 176) and *Cymbopogon jwarancusa* (Jones)Schult. (F = 31% D = 134), were the most dominant species in this site. While highly palatable forage like Aristida funiculate (F = 25% D = 97), *Ochthochloa compressa* (Forssk.) Hilu (F = 22% D = 83), *Launaea resedifolia* L. (F = 17% D = 65) and *Cyperus alopecuroides* Rottb (F = 10% D = 38%) also had representation but displayed lower plant density and little spread in these sites.

Generally, there was a substantial diversity of plant species in this site, but the density and spread of the species were lower compared to the previous two sites (Figure 2). Unpalatable species in this site had minimal representation (Figure 1). Only two unfavorable grazing species were recorded (Table 5). Among these, *Aerva javanica* (Burm.f.) C.Jussieu ex Schult. (F = 26% D = 101) had more representation, but the number of individuals in this site was lowest compared to near and away sites in the study. *Calotropis procera* (Aiton) Dryand. (F = 18% D = 27) was sparsely present in these sites.

**Table 5.** Species Importance Index Values for the site far from human settlements in Rakh Chikkan (R.F: relative frequency, R.D: relative density, R.C: Relative Cover and I.V: Species Importance Value).

| Species | Family | Classification | R.F | R.D | R.C | I.V |
|---|---|---|---|---|---|---|
| *Cenchrus ciliaris* L. | Poaceae | Highly palatable forage | 0.41 | 0.33 | 0.21 | 0.74 |
| *Cynodon dactylon* L. Pers | Poaceae | Palatable forage | 0.34 | 0.58 | 0.17 | 0.92 |
| *Desmostachya bipinnata* (L.) Stapf. | Poaceae | Less palatable forage | 0.01 | 0.01 | 0.04 | 0.03 |
| *Calotropis procera* (Aiton) Dryand. | Apocynaceae | Toxic shrub | 0.02 | 0.009 | 0.07 | 0.03 |
| *Cymbopogon jwarancusa* (Jones)Schult. | Poaceae | Palatable forage | 0.02 | 0.01 | 0.08 | 0.04 |
| *Indigofera hochstetteri* Baker | Fabaceae | Palatable forage | 0.01 | 0.006 | 0.06 | 0.02 |
| *Launaea resedifolia* L. | Asteraceae | Palatable shrub | 0.01 | 0.01 | 0.04 | 0.02 |
| *Citrullus colocynthis* (L.) Schrad | Cucurbitaceae | Palatable (only fruits) | 0.05 | 0.006 | 0.05 | 0.06 |
| *Peganus mharmala* L. | Tetradiclidaceae | Less Palatable herb | 0.02 | 0.009 | 0.06 | 0.03 |
| *Aerva javanica* (Burm.f.) C.Jussieu ex Schult. | Amaranthaceae | Unpalatable Shrub | 0.15 | 0.10 | 0.16 | 0.42 |
| *Aristida funiculate* Trin. & Pupr | Poaeceae | Palatable shrub | 0.10 | 0.09 | 0.08 | 0.29 |
| *Sporobolus arabicus* Boiss. | Poaeceae | Palatable forage | 0.32 | 0.09 | 0.10 | 0.51 |
| *Cyperus alopecuroides* Rottb. | Cyperaceae | Palatable forage | 0.56 | 0.08 | 0.32 | 0.97 |
| *Poa annua* L | Poaceae | Less palatable forage | 0.72 | 0.21 | 0.38 | 1.32 |
| *Schismus arabicus* Nees | Poaceae | Palatable forage | 0.03 | 0.12 | 0.56 | 0.72 |
| *Ochthochloa compressa*( Forssk.) Hilu | Poaceae | Palatable forage | 0.02 | 0.05 | 0.03 | 0.11 |
| *Panicum antidotale* Retz. | Poaceae | Palatable forage | 0.02 | 0.01 | 0.04 | 0.08 |
| *Salsola baryosma* (Roem. & Schult.) Dandy | Amaranthaceae. | Palatable Herb | 0.01 | 0.005 | 0.04 | 0.06 |
| *Lasiurus scindicus* Henrard. | Poaceae | Highly Palatable Forage | 0.03 | 0.003 | 0.05 | 0.03 |

*3.6. Biodiversity Analysis*

Considerable variation in vegetation diversity among the pastoral settlements gradient was observed. The site near pastoral settlements depicted the least biodiversity, as all the diversity indices had low values. Sites away from pastoral settlements were the most diverse. We calculated the Shannon diversity index for analyzing the heterogeneity of the species in the study sites. The results showed that the Shannon index had higher values in the far sites while lowest in the near sites. It was due to a few unpalatable species having vigorous growth spreading in these sites. For analyzing the dominance of a species, the Simpson index was utilized. This usually shows the abundance and dominancy of a species. It was observed that the sites near the pastoral communities had more dominant range vegetation. Sites far from pastoral settlements depicted the lowest value. Buzas and Gibson index was also applied, which displayed more richness and had more evenness compared to near and away sites (Table 6).

**Table 6.** Diversity Indices and their comparison among different study sites.

| Indices | Sites | | |
|---|---|---|---|
| | Near from Community | Away from Community | Far from Community |
| Shannon H | 1.205 | 1.028 | 1.891 |
| Simpson 1-D | 0.5255 | 0.5872 | 0.7023 |
| Menhinick (Spp Richness) | 0.7224 | 0.4697 | 0.5087 |
| Buzas and Gibson's evenness | 0.1971 | 0.2191 | 0.3459 |

*3.7. Biomass Distribution*

A significant relationship was observed between the vegetation and distance from pastoral settlements. According to the Kruskal–Wallis test, the biomass of the vegetation had a significant difference among the distance gradient in all three selected sites ($p = 0.013$), (Table 2). The mean values depicted that as the distance increased, the values of the biomass fall significantly, showing a statistically significant relationship with the distance gradient (Table 1).

The biomass in the current study was measured from various plots in all three sites. The biomass values observed during the samplings were ranging from 5–25 kg/ha. The highest value of biomass was recorded in sites near the human settlements (Figure 3), having a mean value of 18 kg/ha followed by away from settlements (11 kg/ha), and the least biomass values were observed in the locations far from the human settlements (8 kg/ha). The high biomass in the near locations resulted mostly due to the fast growing invasive unpalatable shrub species. While the lower biomass values in the far locations shows sparse availability of fast growing shrubs and more moderately growing palatable herbs.

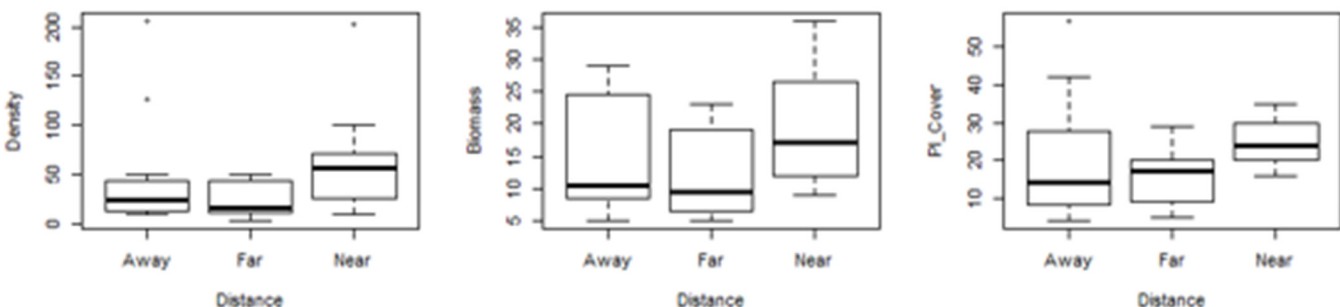

**Figure 3.** Box plots depicting differences in vegetation cover, density and biomass along the settlement gradient.

### 3.8. Vegetation Cover

The results showed a significant relationship of vegetation cover with various sites classified on the distance gradient. The Kruskal–Wallis test identified a statistically significant difference among the vegetation cover and distance from pastoral settlements ($p = 0.003$). The photographs of the vegetation were taken randomly from all the classified sites during the survey. These digital photographs were then used to determine the vegetation cover and bare soil. The results depicted sites near pastoral settlements had the most dominant plant cover compared to all of the allocations in the vegetation survey (Table 1, Figure 4).

As per the results of the vegetation measure, sites near pastoral settlements averaged 65% + 1.33 plant cover. *Aerva javanica*, an invasive and unpalatable species, was the most abundant and had rigorous spread throughout near sites. The sites away from human populations depicted a slight reduction in the plant cover as the values averaged 58% + 3.26 plant cover. The most dominant species in these sites was *Cynodon dactylon*. The sites far from pastoral settlements depicted the least plant cover. The vegetation cover in these locations was a mere 44% + 1.65 cover. These sites, though, had the lowest plant cover but depicted highest diversity among all sites. The most dominant forage species in this site was *Poa annua*. This species was considered as favorable forage species by the livestock in the study area.

Vegetation cover was heavily dependent upon the distance from settlements, as a greater increase in plant cover is displayed from sites away from the human population. This trend is justified by the grazing routes and grazing strategy of the herders. As most pastoral communities in these regions move large distances during monsoon periods to exploit grazing forage away from their settlements while saving the forage near their settlements for harsh drought periods.

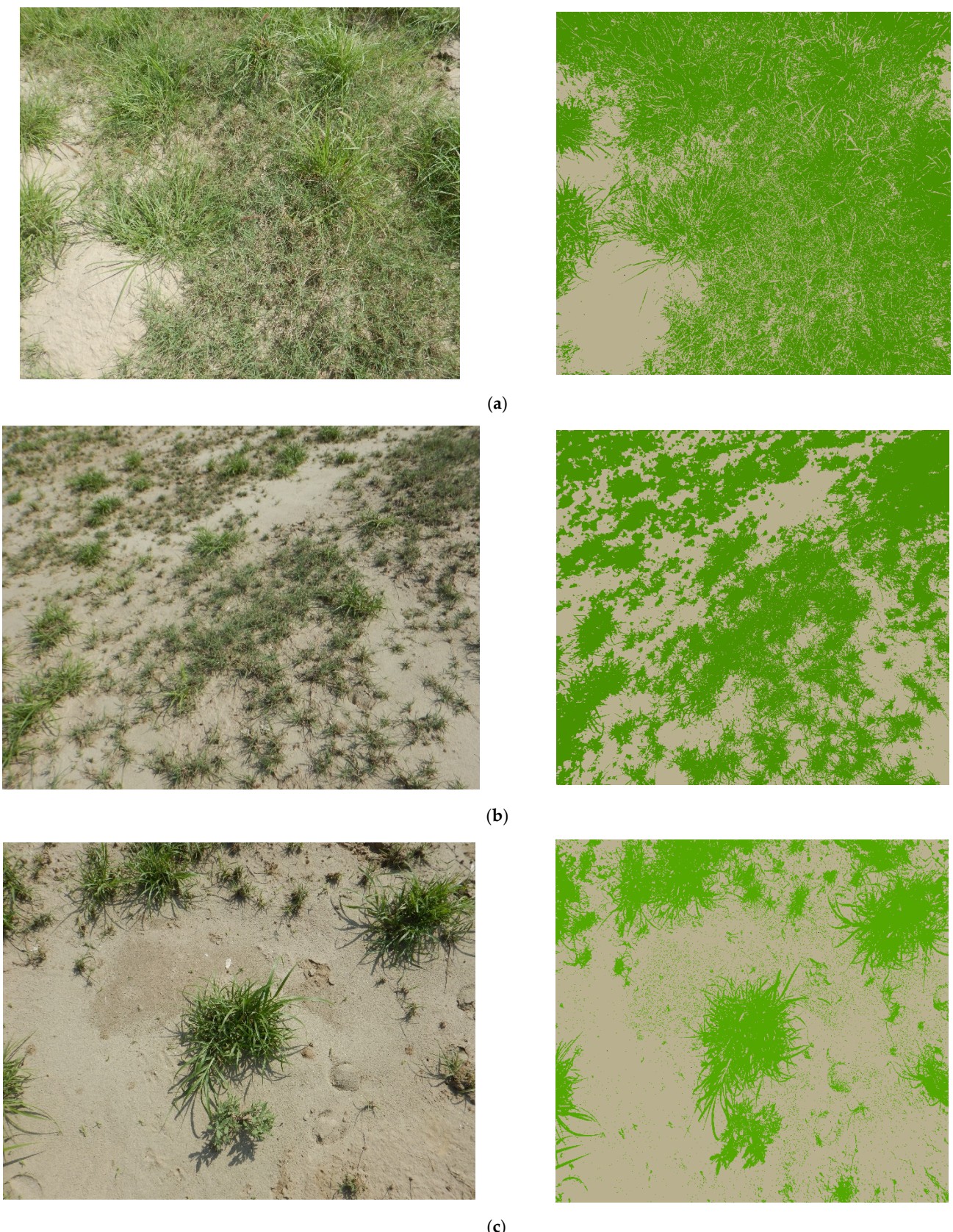

**Figure 4.** (**a**) Site near pastoral settlements (1–2 km) Plant: 64.9% Soil: 35.1 Unclassified: 0; (**b**) Site away from pastoral settlements (2–4 km) Plant: 60.4% Soil: 40.6% Unclassified: 0; (**c**) Site far from pastoral settlements Plant: 29.7% Soil: 70.3% Unclassified: 0.

### 3.9. Nutrient Characteristics

A proximate analysis depicted a significant difference in major and minor nutrients in the selected study sites. According to the Kruskal–Wallis test a significant relationship between nutrients along the pastoral communities' gradient (Table 2). The concentration of nutrients, both trace and macro, tend to decrease as the distance from the population was increased (Tables 1 and 2).

Distance from the pastoral communities significantly decreased micro- and macronutrients values compared to near and away sites (Table 2). It was observed that, comparatively, the highest mean value of both micro- and macronutrients were reordered in forage available in the sites near human settlements (Figure 5). The concentration of the studied macro- and micronutrients (Na, Ca, K) (Fe Zn, Cu, Co and Mn) in the near settlement site was most abundant in *Kochia indica* (354.4, 3822.00, 1654.00 ppm) (64.73, 170.43, 2.45, 3.73, 4.14 ppm). The least values of these nutrients were observed in *Citrullus colocynthis* (88.89, 1327.00, 200.00) (50.18, 86.14, 3.31, 2.23, 1.81).

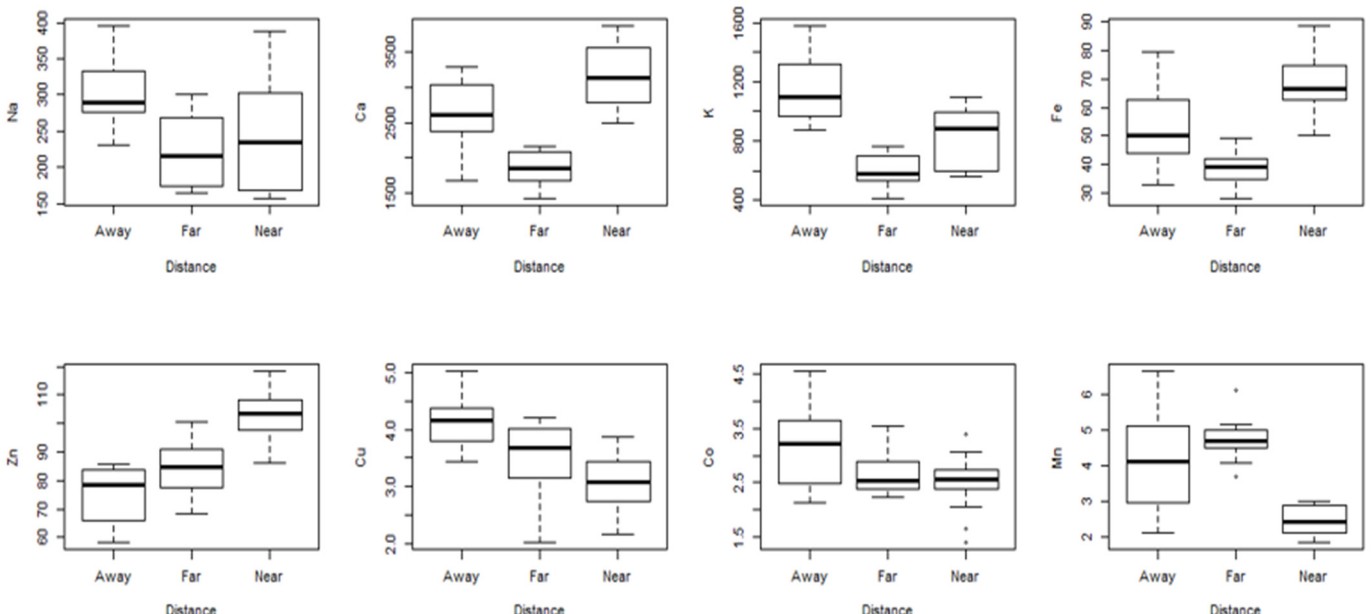

**Figure 5.** Box plots depicting micro- and macro-minerals distribution along the settlement gradient.

Sites away from pastoral settlements showed a significant decrease in the micro- and macronutrients of the forage compared to the near site. Similar to near sites, *Aerva javanica* (375.56, 3197.00, 1174.00) (62.91, 106.86, 4.88, 4.39, 5.14), another unpalatable shrub, showed the highest content of macro- and micronutrients. The lowest concentration of these nutrients were found to be in *Lasiurus scindicus* (112.21, 1198.21, 329.11) (12.87, 115.87, 3.21, 1.21, 5.76). The least concentration of both trace and major nutrients were observed to be in sites far form pastoral settlements. In this location, the highest concentration of nutrients was found in *Cymbopogan jawarncusa* and *Cynodon dactylon* (420.00, 3917.00, 1434.00, 85.64, 112.57, 5.59, 3.75, 7.08). There was a significant difference supported by the Kruskal–Wallis test, depicting a decreasing trend of nutrients' value along the settlements gradient (Table 2).

## 4. Discussion

It is evident from the results that there is a definite impact of pastoral settlements on the structure, characteristics and productivity of the range vegetation. Vegetation diversity tends to increase as we move away from pastoral settlements. Vegetation cover, biomass and nutrient quality showed an inversely proportional association with the pastoral settlements. We saw an increased presence of toxic shrub species in sites near settlements compared

to other sites. Our results highlighted that vegetation characteristics significantly vary with distance from pastoral settlements in arid rangelands of our study region. The study provides baseline information for monitoring trends in range vegetation dynamics and productivity in relation to expanding pastoral settlements.

Most of the plant species recorded during the vegetation inventory were associated with only three families, Poaceae being the most dominant, followed by Amaranthaceae and Cyperacae. Similar trends are observed in most arid lands, where only a few families tend to dominate the landscape. The current study depicted nine families; about 3 of them only had representation of a single species per family. This is consistent with numerous studies that have noted dominance of a few families in most arid rangelands. This is a common feature of desert vegetation, often depicting it an indicator of species adaptation in arid conditions [40].

Plant species having significant grazing value were overall in abundance. An increase in the density and diversity of palatable species was observed along the settlement gradient. Sites near pastoralist population had the least number of desirable species. As distance from the population was increased, the palatable species became comparatively more diverse and slightly dominant. It is in accordance with similar studies implicating an inverse relationship for grazing with palatable species, especially near grazing communities [17,23].

Our results coupled with the findings of comparable studies, depicting a pronounced effect of pastoralist settlements on the composition of range vegetation [41,42]. This showed a decrease in the quantity and quality of nutritional value of palatable forage near its vicinity [43]. A similar study conducted by Hoshino et al. [44] depicted that grazing and a population gradient in a rangeland is a major player governing the vegetation cover, biomass and density. Findings in the present study are in accordance with a research conducted in Botswana, which revealed that increased grazing pressures in active grazing sites, such as tents and water sources, led to a decreased diversity and density of favorable plant species [45].

In the current study, sites near human settlements had high presence of unpalatable and toxic shrubs. These undesirable shrubs are often invasive and have a tendency to spread quickly [46]. Often, many studies have shown unpalatable species rapidly growing in areas with heavy grazing [47]. Palatable species due to overexploitation endures stress thus making way for less utilized unpalatable species to spread quickly.

Our results are in line with Hendricks et al. [48], which implicates that, with an increasing appearance of pastoralist populations, pressure on the grazing resource usually intensifies near the vicinity of the settlements. It has led to a shrinkage of palatable species, thus replaced with inedible and toxic species. A few studies in similar conditions have associated an increasing cover and density of unpalatable species to be an initial indicator of land degradation [49,50].

Pastoral settlements are a key part of the rangeland ecosystem. The feeding pattern, strategy and grazing routes directly affect the plant biomass, cover, diversity and nutritional characteristics [51]. The results showed a decline in biomass values as the distance from the pastoral settlements was increased [52]. Previous studies have shown that areas exposed to overgrazing have experienced a change in species' composition and structure [53,54].

A similar study depicted a decline in biomass and cover of palatable species alongside an increasing livestock population. Unpalatable species like *Kochia prostrata* and *Festuca ovina* firstly decreased, and then in the coming seasons, the growth of these unpalatable species suddenly took an inclining trend [52]. As per these results, our findings also observed similar vegetation composition depicting heavy grazing in these sites. Therefore, these results can deduce its sensitivity to herbivory. Palatable species diminished from near herder's villages and were replaced with rapidly growing unpalatable species with high biomass vegetation cover [53].

Many studies in arid rangelands depict lower vegetation cover and density near to grazers households, camping sites and water sources [55]. It is mainly due to increased dependence upon grazing resources present near the grazers households [56]. The results

in the current study depicts a contradictory trend as the vegetation cover, density and biomass values are higher in sites near to pastoral households as compared to far [57]. This can be explained by the grazing routes and strategy adopted by the herders. An exploratory study conducted in Thal, Pakistan revealed that the pastoralists usually leave some of the grazing resource near to their households. This is done in order to protect these resources for emergency in drought season. In order to save these resources, herders design grazing routes in such a way that they had to travel longer distances from their villages for finding good fodder [51].

Vegetation canopy cover in general showed an inversely proportional relationship to the distance from pastoral settlements. These results are in accordance with various studies depicting lower grazing pressures resulting in with greater distances from the villages [56]. Thus depicting lower vegetation cover as the distance from the villages increased. Further, it is also in line with Jamil et al. [21], which also depicts the vegetation cover decreasing as the distance from the human settlements is increased. Our results are in accordance with a spatio-temporal determination of vegetation canopy cover using remote sensing, in which the authors concluded anthropogenic drivers, mainly distance from rural settlements and livestock density, had a major role to play in governing the vegetation cover of grasslands [58,59].

A mineral analysis of the range species observed in the vegetation survey depicted no clear association with the settlement gradient. It can be justified with the fact that mineral values of the range species are not only dependent upon environmental and anthropogenic factors, but also on the needs of the animal for these elements and their actual absorption potential [60].

Among macro-minerals, sodium, potassium and calcium showed maximum mean and median values in the sites away from pastoral villages while the values of trace minerals showed higher values in the sites far from the pastoral settlements. These high values can be associated with a low grazing pressure, less trampling and reduced anthropogenic activities [61], which is also endorsed by Habib et al. [62], implying that mineral values in grazing sites were usually more compared to protected enclosures.

The study depicted an unpalatable species, such as *Aerva javnica* and *Kochia Indica*, to consist the maximum value of macro-minerals in sites near and away from pastoral settlements while palatable species, such as *Cymbopogan jawarncusa* and *Cynodon dactylon*, displayed the highest values. This implicates lower grazing pressure in the areas away from human settlements, which is also implicated by a similar study [21]. Our results have generally implicated that vegetation dynamics along a settlement gradient have displayed spatial alteration in species' structure and composition. Mainly heterogeneity in livestock distribution, heavy grazing near villages, infrastructure development and unsustainable management can lead towards land degradation.

## 5. Conclusions

The present study aimed to assess the impact of pastoral settlements on the vegetation dynamics and biomass of the range fodder species. We investigated species composition using conventional vegetation inventory coupling Vegmeasure 2 for measuring the canopy cover. The study concluded that increasing pastoral settlements in arid lands is a major factor in changing dynamics of range vegetation.

Vegetation density seemed to decrease along the settlements gradient but the diversity, density cover and biomass of palatable species displayed an increase. This indicated towards high anthropogenic disturbance altering vegetation structure and composition of the grazing forage in these arid lands. The study endorsed the impact of increasing population in range areas to alter species composition and structure due to heavy grazing, agriculture and infrastructure development.

The study displayed the complete dominance of invasive and unpalatable shrub growth near the pastoral villages. Heavy grazing and unsustainable resource management has led to such alteration in species composition and structure. These sites depicted the

dominance of only a few species, increased number of undesirable forage, poor regeneration and lower species diversity. All of these indicators are of human induced land degradation. It poses a serious concern for a sustainable production of forage in these productive rangelands.

The current rangeland monitoring policy in Pakistan usually involves the monthly checking of animal number allotted per pastoral family in the grazing land. Moreover, pastoral communities are prevented from grazing reseeded range plots. Apart from this, no specific and technical range resource monitoring has been carried out by the rangeland department. This study imparts focus on the monitoring in technical terms such as measuring density, biomass, plant cover, nutritional characteristics and palatable plant percentage. Collecting data and monitoring rangelands in scientific terms can a provide better view of the resources and range health. Thus, this study implicates the need for the incorporation of scientific monitoring techniques to the present rangeland monitoring policy.

The present study is important for monitoring the status and dynamics of range vegetation amid expansion of herders' population in order to create a sustainable forage availability. Vegetation in arid rangelands is extremely sensitive due to limited precipitation and severe temperatures. All of these factors make the recovery of plant species in arid environments extremely difficult.

The recovery of plants in such ecosystems may take hundreds of years while the recovery period for a whole ecosystem can take about thousands of years. Thus, the management of such a delicate forage resource for sustaining the livelihood of a large population is of extreme importance. The results of this study can be used as a baseline for developing a monitoring protocol of range vegetation amid expanding populations. Further, the findings could be utilized to detect changes in floral dynamics, establishing habitat protection units and improving efforts for imparting sustainable management plans in arid rangelands.

**Author Contributions:** Conceptualization, A.J. and M.Z.; Methodology, A.J. and M.Z.; Validation, B.A.E.; Formal analysis, A.J.; Data curation, A.J.; Writing – original draft, A.J.; Writing – review & editing, M.Z. and B.A.E.; Supervision, M.Z.; Funding acquisition, B.A.E. All authors have read and agreed to the published version of the manuscript.

**Funding:** This research received no external funding.

**Conflicts of Interest:** The authors declare no conflict of interest.

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
