# Peer review of "Influence of Pastoral Settlements Gradient on Vegetation Dynamics and Nutritional Characteristics in Arid Rangelands"

_sustainability, doi:10.3390/su15064849_

Round 1
Reviewer 1 Report
An interesting, relevant and well conducted research project.
The main questions addressed in the research are vegetation characteristics and the impacts of pastoral settlements. The topic is relevant and adds to knowledge on the topic for policy makers regarding pastoral management and related development financing.
The research is regionally specific and its findings may not have application beyond the region. That said, the methodology that has been developed can be sound basis for further research with similar objectives in other regions with similar characteristics.
The findings are well established and support the conclusions. The report would benefit by a comparison with current policy and development objectives, and proposals for action to address the issues the research presents. Possibly a subject for and extension of the research.
The literature review is extensive, the references are appropriate.
Author Response
Comment: The report would benefit by a comparison with current policy and development objectives, and proposals for action to address the issues the research presents. Possibly a subject for and extension of the research.
Response: Thank You for your well-thought-off comments. As per your suggestion the amendments on line# 568-576 in the conclusion section.
Reviewer 2 Report
Please check the author guidelines regarding references. It is also highly recommended to improve the graphic resolution of the figures and to check text formatting. In addition, "material and methods" should be shortened.

Author Response
- Comment: Can this be quantified? Is there more data available?
Response: This phrase is written in general terms trying to describe the decline of overall productivity of the rangeland due to increasing population. That is why the authors describe no specific quantification.
- Comment: [9, 13, 15], as far as i remember, please check it.
Response: Thanks for highlighting; the authors have corrected the citation style throughout the revised manuscript.
- Comment: Material and methods should be shortened.
Response: As per the reviewer's suggestion, the material and methods are shortened.
- Comment: Correct the units
Response: The units are corrected throughout the manuscript.
- Comment: Figure could be larger and more detailed, e. g.: the error bars, what do they represent? In Addition, it would be better to name the species, if possible
Response: As per the suggestion the species name has been added in the captions and the error bars that represents the uncertainty of a data point is already briefly described in the result section.
Reviewer 3 Report
The manuscript dealt with a relevant topic regarding pastoral settlements in arid rangelands. Quantitative analyses attract even more attention from current scholars; moreover, arid regions and changing ecosystems can be defined better with the results. However, several flaws and issues can be found in the manuscript that must be fixed.
The abstract is too long.
There are several typos, such as sentences in lowercase.
I recommend writing a separate literature-review chapter and separating the introduction from it.
The international relevance should be explained in the introduction as well as in the results. Why are the results important in dry areas of different continents? The originality of the research should also be emphasized more in the conclusion.
The topic concerns population growth, pasture farming, and animal husbandry in dry areas. Furthermore, the authors refer to climate and land use changes and the harmful environmental and ecological consequences of all of these. Since the questions discussed in the introduction are general and voluminous, I think it would be more reasonable to prepare a separate literature review (second chapter). It is worth including wider aspects of the problem, which can increase the interest of the international readership.
The first chapter, as an introduction, could process the literature that discusses vegetation changes. In addition, the authors could elaborate on the research concept and questions that should be more deeply integrated into other research concepts.
I recommend the followings:
Hua, L., & Squires, V. R. (2015). Managing China's pastoral lands: Current problems and future prospects. Land Use Policy, 43, 129-137.
Pricope, N. G., Husak, G., Lopez-Carr, D., Funk, C., & Michaelsen, J. (2013). The climate-population nexus in the East African Horn: Emerging degradation trends in rangeland and pastoral livelihood zones. Global environmental change, 23(6), 1525-1541.
Stige, L. C., Stave, J., Chan, K. S., Ciannelli, L., Pettorelli, N., Glantz, M., ... & Stenseth, N. C. (2006). The effect of climate variation on agro-pastoral production in Africa. Proceedings of the National Academy of Sciences, 103(9), 3049-3053.
Waldron, S., Brown, C., & Longworth, J. (2010). Grassland degradation and livelihoods in China's western pastoral region: A framework for understanding and refining China's recent policy responses. China Agricultural Economic Review.
The novelty of the research should be better placed in an international context. We should see better what kind of work the research was based on.
I think that the existing parts are appropriate, it is only necessary to restructure the first three pages and include new, broader-viewing specialist literature. Of course, this is also necessary for the discussion and conclusions.
For all of this, an overview of the following should be considered:
Kassahun, A., Snyman, H. A., & Smit, G. N. (2008). Impact of rangeland degradation on the pastoral production systems, livelihoods and perceptions of the Somali pastoralists in Eastern Ethiopia. Journal of Arid Environments, 72(7), 1265-1281.
Pei, H., Liu, M., Jia, Y., Zhang, H., Li, Y., & Xiao, Y. (2021). The trend of vegetation greening and its drivers in the Agro-pastoral ecotone of northern China, 2000–2020. Ecological Indicators, 129, 108004
Vuorio, V., Muchiru, A., Reid, R. S., & Ogutu, J. O. (2014). How pastoralism changes savanna vegetation: Impact of old pastoral settlements on plant diversity and abundance in south-western Kenya. Biodiversity and conservation, 23, 3219-3240.
Furthermore, it would be appropriate to display the entire territory of Pakistan in the first figure.
Author Response
Reviewer 3
Comment: The abstract is too long.
Response: As per the reviewer's suggestion the abstract has been shortened.
Comment: There are several typos, such as sentences in lowercase
Response: Thank you for highlighting this important issue. Various typos mistakes are corrected throughout the manuscript.
Comment: I recommend writing a separate literature-review chapter and separating the introduction from it.The international relevance should be explained in the introduction as well as in the results. Why are the results important in dry areas of different continents? The originality of the research should also be emphasized more in the conclusion.
Response: Since as per the the format of the journal the introduction and review should be combined in one chapter as named introduction however the suggestion about international relevance as well as originality has been focused on line # 55-58, 62-66 and 128-130 of the revised manuscript.
Furthermore, it would be appropriate to display the entire territory of Pakistan in the first figure.
Response: As per the suggestion the entire territory of Pakistan is added in the figure.
Reviewer 4 Report
The paper is a an interesting study of impact of pastoral settlement on the environment. The authors used advanced methods of planning and analysis of experimental research material. The experiments are well planned and described in the paper. As for final result, some of them were predictable without research (e.g. Figure 2), but others are not so obvious (point 3.7). I noticed an error in the final paragraph of the sentence "The recovery of plants in such ecosystems may take may take hundreds of of years"
Author Response
1. Comment I noticed an error in the final paragraph of the sentence"The recovery of plants in such ecosystems may take may take hundreds of of years"
Response: The highlighted issue has been corrected on line #588 of the revised manuscript.
Round 2
Reviewer 3 Report
I accept corrections and authors' responses. The article has become more suitable for publication.
Author Response
Thank you.